# Cognitive Dysfunction, an Increasingly Valued Long-Term Impairment in Acromegaly

**DOI:** 10.3390/jcm12062283

**Published:** 2023-03-15

**Authors:** Juan Chen, Zhigao Xiang, Zhuo Zhang, Yan Yang, Kai Shu, Ting Lei

**Affiliations:** 1Department of Neurosurgery, Tongji Hospital, Tongji Medical College, Huazhong University of Science and Technology, Wuhan 430030, China; 2Sino-German Neuro-Oncology Molecular Laboratory, Tongji Hospital, Tongji Medical College, Huazhong University of Science and Technology, Wuhan 430030, China; 3Department of Endocrinology, Tongji Hospital, Tongji Medical College, Huazhong University of Science and Technology, Wuhan 430030, China

**Keywords:** acromegaly, growth hormone, cognition, neuropsychological test, neuropsychological dysfunction

## Abstract

Acromegaly is a chronic disease caused by the overproduction of growth hormone (GH) and accompanying insulin-like growth factor-1 (IGF-1), which is often caused by GH-secreting pituitary adenomas. In addition to its somatic burden, a growing number of studies have found that patients suffering from acromegaly exhibit psychosocial and personality changes. Over the past 70 years, there has been increasing interest in the cognitive impairment and neuropsychological issues of patients with acromegaly, and a variety of neuropsychological and neurophysiological tests have been used to measure cognitive changes in patients. The impact of disease progression status, treatment modalities, and various comorbidities on cognitive function and the mechanisms of cognitive impairment in patients with acromegaly are therefore outlined in this review. Multidisciplinary assessment has important implications for the management of acromegaly, particularly in relation to cognitive function. Here, we summarize the relevant literature concerning cognitive-behavioral research on acromegaly to demonstrate the impact of long-term impairment caused by GH and IGF-1 on the cognitive behavior of patients.

## 1. Introduction

Acromegaly is a chronic disease caused by the increased release of growth hormone (GH) and accompanying insulin-like growth factor I (IGF-1); in most cases, this is induced by a GH-secreting pituitary adenoma and rarely results from ectopic growth-hormone-releasing hormone (GHRH) or GH secretion. Due to the insidious nature of the disease, the clinical features of acromegaly develop chronically so that the average time from the initial onset to diagnosis is typically 4–10 years or longer [1,2]. Progressive somatic disfigurement and extensive systemic clinical manifestations are induced by chronic exposure to excessive growth hormone secretion. The most common complications include diabetes mellitus type 2 (DM2), hypertension, cardiomyopathy, and arthritis, which are responsible for the increased mortality and shorter average lifetime. In addition to the somatic burden, acromegalic patients suffer from psychosocial and personality changes that severely limit their overall quality of life.

Cognitive function is a fundamental part of the psychological condition and crucial for preserving quality of life. Neurocognitive impairment includes the mild dysfunction of attention, memory, executive and learning processes, changes in mood or behavior, and severe Alzheimer’s Disease or other dementias. Hormones from the pituitary gland interact with the brain and can produce profound effects on behavior and cognition. Studies have shown that variations in hormone levels across a person’s life span affect the neurocognitive function [3,4,5,6]. The GH and IGF-1 axis has long been approved for its critical role in brain growth, development, and cognitive function. A significant number of GH and IGF-1 receptors are located in various brain regions, with the GH receptors mainly found in the hippocampus, cerebellum, amygdala, and cerebral cortex, while the IGF-1 receptors are in the amygdala, hippocampus, parahippocampal gyrus, and prefrontal cortex; these are the most critical areas related to executive functions, attention abilities, and memory. Cognitive dysfunction has been confirmed in patients with GH deficiency. Emerging data suggest that GH replacement therapy can play a role in improving cognitive function in these patients.

Various authors have recently shown that acromegaly is associated with alterations in cognitive and neuropsychological functioning. The current review aims to depict the prevalence of cognitive alterations in patients with acromegaly and the associated factors; particular emphasis is given to the effects of sustained GH and IGF-1 excess on the cognitive function of patients with acromegaly and its possible mechanisms.

## 2. Historical Perspective

As early as 1951, Professor Manfred Bleuler [7] reported that some cognitive deficits and personality changes had been observed in patients with acromegalic features. There was a lack of data on the neurocognitive performance of patients with acromegaly at that time. In the 1990s, the Grattan-Smith group [8] designed a retrospective study on the incidence of neuropsychological problems in 38 patients treated with radiation for pituitary adenomas (PA) (among which were 9 acromegalic cases). They found impaired memory and executive functions in patients and did not find a link between this impairment and tumor size, type, or treatment outcome. Although the exact cause was unknown, the authors discovered that cognitive impairment was common in patients with pituitary tumors. Subsequently, the Peace group [9] also reported cognitive impairment exhibited by PA patients, even after other known causes of cognitive dysfunction had been excluded. These cases were not related to emotional disturbances or radiotherapy. In a similar period, Guinan et al. [10] investigated 90 patients treated for PA, 19 of whom were suffering from acromegaly; the study group also showed significant deficits in forward memory as compared to healthy controls. Based on these previous studies, it is clear that patients with acromegaly are at risk of cognitive impairment, although the aspects of the patient’s condition that contribute to cognitive impairment have not been identified.

In 2005, Bonapart et al. [11] first showed that the pessimistic score of acromegalic patients was in the same range as that of patients undergoing active anti-tumor therapy. However, the IGF-1 level of these patients was controlled to stay within the normal range. They proved that the quality of life (QoL) of acromegalic patients was not consistent with the biochemical criteria for disease control, and that it was sometimes even worse. More researchers have realized that evaluating memory, attention, language, and executive functions is intriguing and crucial from the QoL perspective [12,13]. A new study shows that patients’ quality of life and the duration of the disease affect their medication adherence [14]. Meanwhile, the methods of magnetic resonance imaging (MRI) and neurophysiological imaging were employed to find evidence of brain changes in acromegalic patients. In 2009, Tanriverdi et al. [15] found that GH deficiency led to a prolonged P300 event-related potential (ERP) latency, while acromegaly led to a decrease in P300 ERP amplitude. One year later, Leon-Carrion et al. [16] further found abnormal mesencephalic structures on the MRI images of cognitively impaired patients with acromegaly.

In the recent decade, cumulative studies have provided different perspectives for the investigation of cognitive impairment in acromegalic patients at various disease stages—from the naïve status, under therapeutic options, to controlled or even long-term remission [17,18,19]. Nevertheless, conflicting outcomes have emerged regarding cognitive functions. Some studies have suggested that long-term cured acromegalic patients still have attention and memory deficits, while others have shown that they have normal cognition [17,20,21]. Several studies have also investigated the associated or predictive factors and the mechanisms underlying cognitive impairments [18,22,23]. In our paper, we summarize recent advances in the study of the cognitive functional status of patients with acromegaly and their possible mechanisms.

## 3. Cognitive Function Assessment

### 3.1. Cognitive Function Tests

Different tests have been designed to assess cognitive function and to screen mild cognitive impairment in acromegaly. We have summarized all the published neuropsychological tests used to measure the dissociable areas of cognitive impairments (Table 1). As listed below, the most commonly used scales are the Wechsler Adult Intelligence Scale (WAIS), Wechsler Memory Scale (WMS), Beck Depression Inventory (BDI), trail making test (TMT-A&B), Mini-mental-state examination (MMSE), and Montreal Cognitive Assessment (MoCA). Through these tests, clinical neuropsychologists could objectively measure the patients’ performance, including attention and concentration, orientation, short-term and long-term memory, visual and verbal memory, praxis, language, executive function, and decision-making performance. Most of the studies compared the performance of patients with acromegaly to that of healthy subjects, while the other studies recruited patients with non-functioning pituitary adenomas (NFPA) or other chronic diseases as controls [24,25]. The large number of tests used to evaluate functionality as well as the lack of comprehensive, standardized assessment metrics prevented us from accurately measuring the results from different patient subgroups.

### 3.2. Subjective Perception 

Besides the evaluation with designated psychiatrists or physicians, patients’ self-reported evaluation is another important tool for investigating cognitive impairments [20,41]. Yedinak’s team [24] prospectively enrolled 10 active acromegalic patients, 17 controlled acromegalic patients, and 14 NFPA patients, whom they followed over 3 years in a tertiary referral center in the United States. They used questionnaires designed for other diseases, with modifications for acromegaly in order to assess the patients’ self-perceptions of cognitive deficits and their quality of life and health, which included five areas of cognitive function: learning ability, attention and distractibility, mental agility, memory, and verbal memory. As a consequence, they found that the prevalence of cognitive impairment was significantly associated with the patients’ subjective perception, memory, and specific verbal memory. Another study similarly showed that self-perception was affected in acromegaly patients in long-term remission, and that it was strongly associated with the patients’ quality of life [42]. Inevitably, the reliability of self-assessment of cognitive ability does not remain stable throughout the disease as patients’ awareness of their cognitive impairment may decline as it worsens.

### 3.3. Neurophysiological Approach 

The correlation of neurophysiological findings with memory impairment potentiates the importance of cognitive impairment in acromegalic patients. The P300 ERP provides sensitive electrophysiological indicators of task attention and working memory demands and is widely used in assessing cognitive function [43]. Previous studies have shown that the latency of P300 is related to the speed of stimulus assessment, while the amplitude of P300 is related to the update of working memory content [44,45]. Tanriverdi’s study [15] showed reduced P300 amplitude in patients with acromegaly and provided electrophysiological evidence of reduced working memory in GH overload, which laid the foundation for electrophysiological studies of cognitive impairment in patients with acromegaly. There was a significant negative correlation between P300 amplitude and temporoparietal bone thickness in healthy subjects [46], implying that changes in P300 in patients with acromegaly may be associated with changes in the cranial structure due to high GH. Another study using an electroencephalogram (EEG) and tomographic MRI found that patients with active acromegaly had reduced EEG activity in the brain regions closely associated with memory, namely the prefrontal and middle temporal cortex [16].

## 4. Cognitive Impairments in Acromegaly

In two studies with a heterogeneous population of acromegaly patients, the prevalence rate of cognitive impairment varied between 10 and 72% [19,27]. In a cross-sectional study of 55 patients with acromegaly [27], Sievers et al. found that cognitive impairment was prevalent in patients, with 67.3% of patients not meeting the critical levels on at least one test; more specifically, up to 33.3% had impaired attention, 24.1% had memory impairment, and 16.7% had poor executive function performance. Furthermore, the Brummelman group reported that up to 72% of the patients they studied showed low scores on at least one test, while no association was found with the patients’ treatment regimen and biochemical control status [28].

Memory assessment revealed significant differences in most studies. Lon-Carrion et al. [16] found that patients with active untreated acromegaly showed a severe impairment of memory in both the short and long terms, and that the degree of impairment was significantly associated with GH and IGF-1 hyperactivity. In addition, patients had less EEG activation in areas closely associated with memory, such as the prefrontal and middle temporal cortices. Furthermore, Crespo’s team [29] found a strong correlation between memory performance and anxiety and depressive symptoms in patients with acromegaly; they suggested the possibility of emotional support to improve cognition.

Executive function is one of the most critical and complex functions in neurocognitive processing, including working memory, flexibility, reasoning, planning, execution, inhibitory control, and problem-solving [47]. This denotes the ability to avoid adverse stimuli and use past experience to guide current actions, and its damage is often imperceptible. Decreased executive ability in acromegalic patients has been found in several studies [16,31,33]. Müssig et al. [48] reported the risk of attention and working memory impairment in patients operated on for pituitary adenoma (including 13 patients with acromegaly), which was not observed in patients with other chronic diseases. In addition, these patients with acromegaly were themselves aware of deficits in executive function, including executive processing, inhibitory control, and working memory. High levels of IGF-1 and the duration of the disease may be responsible for impairment in specific aspects of executive function [31]. Reducing difficulties in the executive syndrome has important implications for improving the QoL of patients.

To date, the largest investigation of cognitive function in acromegaly that has been carried out involved 223 patients from five referral centers in Italy [19]. All the patients had undergone surgery or were under medical treatment. The results showed that cognitive function was affected in about 10% of the patients, and interestingly, female patients seemed to show more visuospatial and verbal working memory problems. However, beyond that, male patients had more significant impairment. Combined with Acromegaly QoL Questionnaire (AcroQoL) scores, GH levels, comorbidities, disease duration, and patient age are among the many factors that may negatively impact patients’ cognitive function and quality of life [49].

## 5. Correlation Factors

### 5.1. Various Disease Statuses 

Over the past few decades, several studies have investigated the cognitive status of acromegalic patients, with inconsistent results.

Patients with acromegaly may have cognitive dysfunction at every stage of the disease, from pre-diagnosis to post-surgery. Mild-to-moderate cognitive impairment exhibited by patients with untreated active acromegaly may be associated with prolonged exposure to excess GH and IGF-1 [16]. Cognitive impairment may persist even though surgical and medical treatments can directly or indirectly improve brain function [50]. In a cross-sectional study of 50 patients with acromegaly [28], patients performed lower on cognitive tests than the healthy controls. Nevertheless, there were no significant differences among active acromegaly patients, acromegaly patients in remission, and NFPA patients. Additional studies likewise suggest that cured acromegaly patients did not achieve better cognitive outcomes compared to patients with uncontrolled hormone production [24,51], implying that long-term exposure to excess GH and IGF-1 may have long-term effects on brain function.

On the contrary, other studies have reported that these cognitive disorders may improve after controlling GH and IGF-1 hypersecretion. Tiemensma et al. [20] first reported that compared to matched non-functioning pituitary macroadenoma (NFMA) patients and healthy controls, there were no significant differences in overall cognitive function in patients with long-term cured acromegaly. Although in this study, the patients with acromegaly performed poorly on executive and verbal memory. However, the endocrinological status of the disease was not a significant predictor of better neurocognitive performance. Another group suggested that acromegaly has no long-lasting negative impact on attentional, memory, and psychomotor functions as they found no remarkable cognitive decline when comparing the under-controlled group with the remission group [21]. A possible explanation for this finding might be the clearly increased oversecretion of GH even in patients without remission as compared to their pre-operative status. Martín-Rodríguez et al. [17] also demonstrated that the longer the period of biochemical remission after acromegaly, the better the neurocognitive status of the patients.

### 5.2. Treatment Modalities 

Acromegalic patients may need multimodality treatments to reach remission, which include surgical adenectomy, medical treatment, radiotherapy, or a combination of these treatments. The therapeutic efficacy and impact of these treatment options on cognitive function has been controversial. As early as 1997, memory and executive function impairments were reported in patients with pituitary tumors who had undergone surgery [9]. These impairments in neurocognitive function were independent of the type of surgery and access. To investigate the effects of treatment modalities on memory and general intellectual function (IQ), Guinan et al. [10] also investigated 90 patients who had been treated for pituitary adenomas, 19 of whom were suffering from acromegaly. All their treatment groups showed forward memory deficits as compared to the healthy controls, suggesting that the treatment and not the tumor caused these deficits. Interestingly, there was no significant difference between the cognitive outcomes of surgery and radiation therapy. The patients with more severe memory deficits had received adjuvant radiotherapy, and postoperative MRI suggested that their diencephalic structures were damaged. Another study concluded that patients who had undergone successful pituitary adenoma surgery exhibited deficits in cognitive function, particularly attention and cognitive function, reinforcing the correlation between treatment and functional impairment [48]. To date, there is no specific study on patients with acromegaly that investigates the correlation between surgical intervention and cognitive function by comparing acromegalic patients with or without surgery.

Pharmacological treatment is indicated for patients who could not receive surgery or are at risk of surgery. Acromegalic patients receiving long-term, long-acting somatostatin analogue therapy were reported to have lower learning, attention, and planning scores than healthy controls with a match for age, gender, and level of education [18]. To assess the influence on cognition of prolonged medical treatment after an initial transsphenoidal surgery, Brummelman et al. [28] evaluated the cognitive status of patients controlled by medication; after a comparison with the surgical remission group, they concluded that the existing medical treatment for GH excess did not affect memory and executive function. A pilot study also concluded that cognitive function and cortical thickness in patients with acromegaly are unaffected by medication and glycemic control [23]. Curiously, unlike the somatostatin receptor ligand, there was no significant association between executive function and thalamocortical thickness in the Pegvisomant-treated patients. Poor thalamocortical connectivity may play a central role in the potential neuropathology of type 2 diabetes-related cognitive dysfunction [52]. Thus, Pegvisomant may protect thalamic nerves, but more in-depth studies are needed to demonstrate this.

Radiotherapy may not cause the impairment of cognitive function. Gamma Knife radiosurgery (GK) is an important modality for controlling tumor regrowth and GH-oversecretion in acromegaly. Fewer data are available on its long-term side effects, especially on cognitive function. Lecumberri [30] reported that postoperative two-field conventional radiotherapy was independently associated with impairments in verbal memory and executive function. In a later study, Castinetti et al. [37] showed that out of 64 patients with acromegaly, there were 27 who had received radiation therapy and demonstrated a neurocognitive function that was not significantly different from that of the unexposed group, although up to 23.8% of the patients performed poorly on at least one test.

### 5.3. Patient’s Age 

Acromegaly may impair cognitive functions in the geriatric population. Hatipoglu et al. [22] showed that MMSE scores were significantly lower and dementia was more common in elderly patients with acromegaly than in matched controls. This implies that acromegaly poses a greater risk for cognitive function and daily living as well as heightens the possibility of malnutrition in older adults. Pivonello et al. [19] also showed that disease duration and patient age were among the main determinants of these mental force conditions, with worsening cognitive symptoms for patients with a disease duration of more than 10 years and older patients. Likewise, slightly younger patients have been found to commonly have problems with visuospatial and verbal memory. Wennberg et al. [32] showed that patients with an average age of 56 are also more likely to develop cognitive impairment than the general public.

### 5.4. Other Comorbidities 

Acromegaly can cause systemic comorbidities due to the overproduction of pituitary hormones, and a positive correlation has been observed between these comorbidities and cognitive impairments in acromegalic patients. A larger multicenter series showed that the prevalence of arterial hypertension reached about 30% in patients with acromegaly, which was as high as the comorbidity for diabetes [53,54,55,56]. Among the comorbidities, patients with acromegaly accompanied by diabetes have been observed to perform poorly in abstract thinking, short-term memory, and executive function, while patients with hypertension perform poorly in executive function [18]. However, there is a lack of association between cortical thickness and cognitive dysfunction in patients with glycemic control [23].

In addition, patients with comorbid obstructive sleep apnea syndrome have exhibited significantly impaired attention [19]. Wennberg et al. [32] found an association between sleep quality, somnolence, and cognitive performance in 67 patients with acromegaly. Overall, 6–10% of the patients showed impairment in cognitive function. Their results demonstrate that poorer sleep quality is associated with quality of life, thus showing a correlation with abnormal cognitive function.

There also is a link between patients’ anxiety and depressive symptoms and their memory and decision-making abilities. Crespo et al. found that emotional support improved the patients’ cognition, suggesting that improvements in anxiety and depression might help [29].

## 6. Mechanism of Excess GH and IGF-1 in Cognitive Impairments

Pituitary hormones and their downstream hormones have important effects on cognitive function. For example, thyroxine promotes the formation and maturation of the central nervous system [57], while Cushing disease (CD) causes cognitive dysfunction in children [58]. In physiological functions, GH plays a vital role in hippocampal development, neurogenesis, and cognitive function. GH induces parvocellular hippocampal interneurons to express the phosphorylated form of the signal transducer and activator of transcription 5 (pSTAT5) [59]. Cells containing γ-aminobutyric acid (GABA), growth inhibitors, and neuropeptide Y are involved in hippocampal development and cognitive regulation [60]. The growth hormone receptor (GHR) and IGF-1 activate N-methyl-D-aspartate (NMDA) receptors to enhance long-term potentiation (LTP) and improve memory via the cyclic adenosine monophosphate-protein kinase A (cAMP-PKA) pathway [6]. Due to these critical roles of GH in neurological development, cognitive impairment has been observed in patients with growth hormone deficiency (GHD) and has been clinically confirmed [61,62]. Figure 1 summarizes some pathways through which the growth hormone acts on hippocampal and cognitive functions. Regression analysis showed that the longer the duration of untreated acromegaly, the more likely it was to have neurocognitive functional complications; however, an association between neurocognitive function and GH was only found in naïve patients [17]. Nevertheless, there is a lack of studies dedicated to the effects of the excessive growth hormone on hippocampal and cognitive functions. Therefore, the pathophysiological mechanism of cognitive dysfunction in acromegaly requires more in-depth basic research.

Blood ghrelin has been found to be lower in patients with acromegaly than in healthy controls and non-functioning pituitary adenomas, and a lower growth hormone level after surgery causes ghrelin to rebound [63,64,65]. Growth hormone secretagogue receptor 1a (GHSR1a) co-expressed with dopamine D1 receptor (DRD1) in the hippocampus [66], which activates Ca^2+^-calmodulin-dependent kinase II (CaMKII), glutamate receptor exocytosis, synaptic reorganization, and the expression of early markers of hippocampal synaptic plasticity, is critical for DRD1 to initiate hippocampal synaptic plasticity, while GHSR1a inactivation inhibits DRD1-mediated hippocampal behavior and memory. Furthermore, ghrelin plays a vital role in the progression of Alzheimer’s disease [67,68] as well as in amyloid β homeostasis and tau protein hyperphosphorylation [69,70]. Thus, ghrelin abnormalities may play a role in the cognitive impairment of patients with acromegaly.

The MRI results of patients with acromegaly suggest possible structural alterations. Sievers et al. [27] found increased gray and white matter in the cerebral hemispheres, particularly in the left and right lateral hippocampus, of patients with acromegaly, especially early in the course of the disease. This finding was confirmed by Yuan et al. [71] in 2020, further finding a decrease in cerebrospinal fluid volume (CSFV) and that these changes significantly correlated with serum GH and IGF-1 levels. A subsequent in-depth study by Yuan et al. [36] showed a general decrease in T1/T2 weighted ratio in most cortical and white matter in patients with acromegaly, implying extensive cortical and white matter demyelination and myelin regeneration. Alterations in structures such as the cerebral cortex, gray matter, and white matter may be responsible for neuropsychological dysfunction, including cognitive impairment, in patients. Unfortunately, patients’ cognitive changes have not been linked to structural changes seen on MRI, so it is still challenging to use anatomical changes as a reliable predictor of cognitive dysfunction.

## 7. Conclusions

In conclusion, cognitive impairment is an essential comorbidity of acromegaly that requires more attention. Long-term acromegaly leads to cognitive dysfunction in patients, with disease status and age as influencing factors. The mechanism of cognitive impairment in patients may result from a combination of factors in which GH and IGF-1 play an essential role. Future research is needed to further investigate the mechanisms of its occurrence, define appropriate multidisciplinary assessment protocols, and find therapeutic measures to mitigate this impairment.

## Figures and Tables

**Figure 1 jcm-12-02283-f001:**
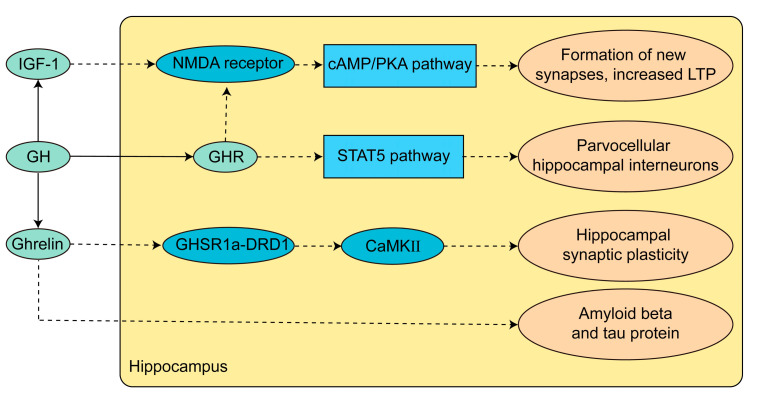
Pattern of GH and IGF-1 action in the hippocampus. GH acts on parvocellular hippocampal interneurons by activating the STAT5 pathway in order to regulate hippocampal development. GH and IGF-1 activate the cAMP-PKA pathway through NMDA receptors to enhance LTP. Excess GH negative feedback decreases ghrelin and affects synaptic plasticity in hippocampal dopamine neurons. Ghrelin has also been implicated in the development of Alzheimer’s disease. Solid arrows indicate a direct interaction, dashed arrows indicate an indirect interaction. Abbreviations: IGF-1, insulin-like growth factor-1; NMDA, N-methyl-D-aspartate; cAMP, cyclic adenosine monophosphate; PKA, protein kinase A; LTP, long-term potentiation; GH, growth hormone; GHR, growth hormone receptor; STAT5, signal transducer and activator of transcription 5; GHSR1a, growth hormone secretagogue receptor 1a; DRD1, dopamine receptor-1; CaMKII, Ca^2+^-calmodulin-dependent kinase II.

**Table 1 jcm-12-02283-t001:** Neuropsychological tests to assess cognitive function in patients with acromegaly.

Reference	Year of Publication	N	Treatment Beyond Surgery ^1^	Tests	Effective Results ^2^
Sonino et al. [26]	1999	10	Lanreotide	SSQ, SLPP	+
Tanriverdi et al. [15]	2009	18	−	P300 ERP	+
Leon-Carrion et al. [16]	2010	16	−	TMT-A, TMT-B, LMW-R, BDI-II, AcroQoL	+
Tiemensma et al. [20]	2010	68	SRL, PEG	MMSE, WMS	+
Psaras et al. [21]	2011	37	−	AcroQoL, WAIS-R, TMT-A, SF-36, SCL-90-R	+
Sievers et al. [27]	2012	55	Radiotherapy	TAP, WMS, VLMT, SPM, RWT	33.3%
Brummelman et al. [28]	2012	50	SRL, PEG	15 Words Test, Ruff Figural Fluency Test	+
Martín-Rodríguez et al. [17]	2013	102	−	LMW-R, DST, TMT-B, SCWT	+
Yedinak et al. [24]	2014	27	SRL	FACT-Cog	+
Hatipoglu et al. [22]	2015	30	SRL	MMSE, AcroQoL, GDS	+
Crespo et al. [29]	2015	31	SRL, Radiotherapy	IGT, RAVLT, BDI-II	+
Lecumberri et al. [30]	2015	124	Radiotherapy	MMSE, BVRT, WCST, BT	+
Alibas et al. [18]	2017	42	SRL, Radiotherapy	OVMS, WAIS-R, TMT-A, TMT-B, SCWT, BDI	+
Shan et al. [31]	2017	42	−	Stroop Test, Verbal Fluency Test, HSCT, SART	+
Dimopoulou et al. [25]	2017	81	−	Testing in the domains of attention, memory, and executive functions	+
Wennberg et al. [32]	2019	67	SRL, PEG	SSRT, CBTT, ROCF, TMT-A, TMT-B	6–10%
Solomon et al. [33]	2019	19	−	AcroQoL, TMT-A, TMT-B, Stroop Test	+
Kunzler et al. [34]	2019	23	SRL	SF-36, BDI	+
García-Casares et al. [23]	2021	23	−	MMSE, CWST, WAIS-III, WMS-III, BDI-II, TMT-A, AcroQoL	-
Gagliardi et al. [35]	2021	42	SRL, PEG, DA	SF-36, AcroQoL	+
Yuan et al. [36]	2021	29	−	MoCA, BDI, DSST, SAS	+
Castinetti et al. [37]	2021	64	Radiotherapy	Grober and Buschke Test, STMT; PASAT	−
Hatipoglu et al. [38]	2022	33	SRL, Radiotherapy	BDI, DST, DBT, MBT, WAIS-R, BNTSF	−
Pivonello et al. [19]	2022	223	SRL, PEG, CAB	BDI-II, STAI, CBTT, TMT-A, TMT-B	10%
Xie et al. [39]	2022	55	−	MoCA	+
Kan et al. [40]	2022	58	Radiotherapy	BDI, Rosenberg Self-esteem Scale	+

^1^: “−” means no explicit description. ^2^: Rate of cognitive impairment in patients with acromegaly. “+” means the rate of cognitive impairment in patients with acromegaly is higher than in controls; no specific prevalence described. “−” means no differences detected between the acromegaly patient group and other groups. The most commonly used scales in the literature are the Wechsler Adult Intelligence Scale (WAIS), Wechsler Memory Scale (WMS), Beck Depression Inventory (BDI), trail making test (TMT-A&B), Mini-mental-state examination (MMSE), and Montreal Cognitive Assessment (MoCA). A total of 23 out of 26 studies showed statistical differences in cognitive function. Abbreviations: SSQ, Social Situation Questionnaire; SLPP, Screening List for Psychosocial Problems; P300 ERP, P300 event-related potential; TMT-A, TMT-B, trail making test; LMW-R, Luria’s Memory Words Test-Revised; BDI, Beck Depression Inventory; AcroQoL, Acromegaly Quality of Life Questionnaire; SRL, somatostatin receptor ligand; PEG, pegvisomant; MMSE, Mini-Mental-State Examination; WMS, Wechsler Memory Scale; WAIS, Wechsler Adult Intelligence Scale; SF-36, 36-Item Short Form Health Survey; SCL-90-R, Symptom Checklist 90-Revised; TAP, estbatterie zur Aufmerksamkeitsprufung; VLMT, Verbaler Lern- und Merkfahigkeits Test; SPM, Raven Standard Progressive Matrices Test; RWT, Regensburger Wortflussigkeits-Test; DST, Digit Span Test; SCWT, Stroop Color and Word Test; FACT-Cog, The Functional Assessment of Cancer Therapy Cognitive Scale; GDS, geriatric depression scale; IGT, Iowa Gambling Task; RAVLT, Rey Auditory Verbal Learning Test; BVRT, Benton Visual Retention Test; WCST, Wisconsin Card Sorting Test; BT, Barcelona Test; OVMS, Oktem Verbal Memory Scale; HSCT, Hayling Sentence Completion Test; SART, Sustained Attention to Response Task; SSRT, Short Story Recall Test; CBTT, Corsi Block-Tapping Test; ROCF, The Rey–Osterrieth Complex Figure Test; CWST, Color-Word Stroop Test; DA, dopamine agonists; MoCA, Montreal Cognitive Assessment; DSST, Digit Symbol Substitution Test; SAS, Self-Rating Anxiety Scale; STMT, Stroop and Trail Making Test; PASAT, Paced Auditory Serial Attention Test; DBT and MBT, Days and Months Backwards Test; BNTSF, Boston Naming Test Short Form; CAB, cabergoline; STAI, State-Trait Anxiety Inventory.

## Data Availability

The data presented in this study are available from the corresponding author upon reasonable request.

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
