# Peer review of "Cognitive Dysfunction, an Increasingly Valued Long-Term Impairment in Acromegaly"

_jcm, 2023, doi:10.3390/jcm12062283_

Round 1

Reviewer 1 Report

Dear authors,

You have written a comprehensive literature overview, but its present form is difficult to follow. Your work would lead to a better scientific contribution if done according to the PRISMA statement as a systematic review. Please include methods of how studies were selected, define all variables for which data were sought, and describe methods used for assessing the risk of bias in individual studies and results accordingly. 

It is a valuable contribution to this field and would improve your work.

Author Response

Thank you for your valuable comment. We have carefully considered the comment and have revised the manuscript accordingly. Please find below our response to the reviewer‘s comment.

Point 1: You have written a comprehensive literature overview, but its present form is difficult to follow. Your work would lead to a better scientific contribution if done according to the PRISMA statement as a systematic review. Please include methods of how studies were selected, define all variables for which data were sought, and describe methods used for assessing the risk of bias in individual studies and results accordingly.

Response 1: Thank you for this valuable feedback. We have specified the strategy of the literature search and the selection criteria. All published English-language literature that met the criteria totaled 26. There were significant differences among all studies in the evaluation criteria used, the type of patients with acromegaly, the control group, and the study format. The differences would make the bias between studies extensive and challenging to eliminate, making it difficult to conduct a systematic review based on the PRISMA statement. Therefore, we only list the data and do not analyze them in depth. We recommend the use of more uniform tests of cognitive function in patients. Your suggestions have been constructive, and we sincerely thank you!

Method

Search strategy: A search for published articles (up to 31 December 2022) was conducted in Medline and Web of Science. “Cognition”, “cognitive function”, “cognitive dysfunction”, “memory”, “attention”, “executive function”, and “quality of life” were used to retrieve articles on cognitive dysfunction. All these keywords were combined with “acromegaly”, “pituitary adenoma”, “gi-gantism”, and “growth hormone”.

Selection criteria: Only peer-reviewed clinical studies published in English were included in this review. The following inclusion criteria were used:

1. Patients had been identified as patients with acromegaly, both primary and treated.

2. Cognitive function of patients with acromegaly had been evaluated.

The exclusion criteria were the following: Cognitive function of patients with acromegaly had not been evaluated. Case reports were excluded. Each study had to specify the year of publication, number of patient cases, treatment protocols, assessment methods, and results. All clinical studies were retrospective, lacked randomized controlled trials, and had different treatment protocols and assessment criteria. As a result, overall bias was difficult to control, so no systematic evaluation was performed.

Reviewer 2 Report

This is a comprehensive review on cognitive dysfunction in acromegaly. It is well written, easily red and understandable. 

I would suggest to further clarify legend in the Table 1, since I'm not quite sure  what "% of patients, "+" means efective, "-" means no difference" means, compared to what? 

Also, I would suggest to insert Figure explaining pathophysiological mechanisms of excess GH and IGF-1 on development of cognitive impairment. 

Reviewer 3 Report

Thank you for the opportunity to review the manuscript “Cognitive dysfunction, an increasingly valued long-term impairment in acromegaly” by Juan Chen et al. Management of acromegaly is often challenging and therefore a review regarding the neurocognitive impact of both the disease and the treatment is valuable.

Please see line 36: “The most common complications include type II diabetes, hypertension, cardiomyopathy, arthritis, et al.,”- rather use diabetes mellitus type 2 (DM2); “et al.”- unclear, etc.?- I would avoid using it.

I would recommend referring to the latest study by Bulgarian group published this year: Pharmaceutics 2023, 15(2), 438; https://doi.org/10.3390/pharmaceutics15020438

Author Response

We are very grateful to the reviewers for their valuable suggestions, which allowed us to improve the quality of the manuscript. Every revision suggestion and comment made by the reviewers have been accurately incorporated and considered. The following is a point-by-point response to the reviewers' comments.

Point 1: Please see line 36: “The most common complications include type II diabetes, hypertension, cardiomyopathy, arthritis, et al.,”- rather use diabetes mellitus type 2 (DM2); “et al.”- unclear, etc.?- I would avoid using it.

Response 1: As suggested by the reviewer, we used ignored the correct writing of nouns and made redundant descriptions. We have revised the writing of diabetes mellitus type 2 in the text and deleted "et al."

Point 2: I would recommend referring to the latest study by Bulgarian group published this year: Pharmaceutics 2023, 15(2), 438; https://doi.org/10.3390/pharmaceutics15020438

Response 2: Thank you for this valuable feedback. This paper provides high-quality results as a prospective clinical study. It draws an important conclusion: medication adherence in patients with acromegaly is correlated with their quality of life. This implies that alterations in cognitive function and quality of life in patients with acromegaly may influence the treatment of the disease. We have now cited this paper as “Reference 14” in the “Historical Perspective” part. We sincerely thank you for your comments

Reviewer 4 Report

The manuscript „Cognitive dysfunction, an increasingly valued long-term impairment in acromegaly“ provides a comprehensive overview of the cognitive impairment associated with acromegaly. As is widely known, the chronic exposure to excessive growth hormone results in progressive physical disfigurement and various systemic complications, but as the authors point out, it also leads to psychosocial and personality changes, which can negatively impact the quality of life, as well as impairment of cognitive function, including attention, memory, executive function, and learning. The authors lay out fundamentals of early research in this domain since the 1950s, which only gained traction in the 1990s when researchers started to investigate this issue in depth. Various studies using magnetic resonance imaging (MRI) and neurophysiological methods have been conducted in recent years to understand the cognitive impairment of acromegalic patients and the possible mechanisms behind it, but with conflicting results.

The article summarizes the different tests used to assess cognitive function in acromegalic patients and the results obtained, highlighting the conflicting results of various studies. While some studies showed that long-term cured acromegalic patients still have attention and memory deficits, others suggest normal cognition.

In conclusion, the article highlights the need for further research to better understand the cognitive impairment associated with acromegaly and to develop effective treatments to improve cognition and therefore quality of life of affected patients. The literature presented is relevant and seems comprehensive.

The following is a list of points that could be improved in the article:

* The language is sometimes odd: especially the wording and occasionally the syntax reveal the non-native-speaking origin of the authors. Throughout the article, there are instances of inconsistent or incorrect usage of punctuation. I recommend to let a native-speaking colleague (or medical professional / biologist) to revise and improve the manuscript.

* The section on the "Historical Perspective" could benefit from a clearer and more concise description of the key findings from each study mentioned.
* In the section on "Cognitive Function," there could be a clearer explanation of the relationship between hormones from the pituitary and their effects on behavior and cognition.
* The section on "Cognitive Function" could also benefit from a clearer explanation of the receptors for GH and IGF-1 and their role in the brain, thus extending the scope of the review from clinical phenomenology to basic research on cerebral GH/IGF-1 biology. It is not strictly necessary, but it would provide additional value.
* In the section on "Cognitive Function," there could be a clearer explanation of the connection between GH deficiency and cognitive function and how GH replacement therapy can play a role in improving cognitive function.

* Authors' last names should suffice in sentences such as "the Grattan-Smith PJ group [...]"; in the table omit the first name initial.
